# COMPARISON OF NEURAL NETWORK ARCHITECTURES IN THE THERMAL EXPLOSION APPROXIMATION PROBLEM

## ABSTRACT

This study investigates the effect of neural network architecture on the accuracy of data-driven modeling of thermal explosions in a hydrogen–oxygen–air mixture. Using a reduced kinetic mechanism for 11 reagents, the thermal explosion process is simulated under specified initial pressure and temperature conditions, generating time-resolved data. We compare three architectures: a standard multilayer perceptron (MLP), a DeepONet–inspired model, and our U-Net–style residual network, evaluating their ability to capture transient dynamics and key reaction regimes. Our results demonstrate that network architecture has an important impact on predictive performance. The U-Net architecture consistently outperformed the other models, achieving a mean squared error (MSE) of 0.0013 with a standard deviation (STD) of 0.0218, demonstrating high fidelity in capturing both rapid transients and slower reaction dynamics. In contrast, the DeepONet-inspired model and the MLP achieved MSEs of 0.0181 (STD 0.0581) and 0.0202 (STD 0.0682), respectively, indicating reduced accuracy and greater variability in predictions. The large spread in error is due to the fact that neural networks are not always able to accurately approximate the various modes of the combustion process. Despite testing various architectures and using a fairly large dataset, the problem remains unresolved. These findings indicate the importance of selecting appropriate network architectures to combine deep learning with chemically detailed kinetic modeling. Such careful selection open the way for more reliable and interpretable predictive models in combustion and reactive-flow applications.

## 1 INTRODUCTION

The numerical simulation of gas-dynamic processes in high-energy systems, such as engines, gas turbines, and other propulsion or power-generation devices, requires a detailed description of complex physicochemical phenomena. These include turbulent combustion, shock wave interactions, and multiphase flows, all of which are strongly influenced by chemical kinetics. Modern computational fluid dynamics (CFD) approaches employ high-fidelity reaction mechanisms that account for hundreds or even thousands of elementary chemical reactions among dozens of interacting species. Such detailed models are essential for accurately predicting critical processes like ignition delay, flame stabilization, deflagration-to-detonation transition (DDT), and detonation wave propagation.

However, the high level of detail in these simulations comes at a considerable computational cost. Resolving all relevant spatial and temporal scales in 3D simulations demands immense computational resources, making such calculations impractical for routine engineering analyses or real-time applications. To address this challenge, researchers employ various acceleration techniques, including reduced-order modeling, adaptive mesh refinement, and hybrid combustion models that balance accuracy and efficiency. Despite these advancements, the trade-off between computational feasibility and predictive accuracy remains a key research focus in computational combustion and reactive gas dynamics (Pantano et al., 2004; Smirnov et al., 2015).

The main computational bottleneck in coupling hydrodynamics with chemical kinetics lies in solving stiff systems of ordinary differential equations (ODEs) that describe the temporal evolution of temperature and species concentrations.

Despite advances in mechanism reduction techniques (Løvås, 2009; Koniavitis et al., 2016; Lu et al., 2021) and tabular approaches (Pope, 1997), the computational cost of detailed chemical kinetics still constitutes a substantial fraction of the total simulation time. In recent years, machine learning techniques, in particular neural network models, have attracted growing interest as a means to accelerate reactive flow simulations. Training a neural network to approximate the thermochemical evolution of a reactive medium can reduce computational demands while maintaining acceptable accuracy (Tonse et al., 1999; Ji & Deng, 2021).

In this context, the selection of neural network architecture plays a decisive role in achieving an appropriate balance between predictive accuracy, generalization capability, and computational efficiency. A growing body of research indicates that architectural features—such as network depth, width, residual connections, and hierarchical representations—exert a profound influence on both the stability of predictions. For chemically reactive systems, architectures capable of capturing multiscale dependencies are particularly critical, as the temporal evolution of species involves both ultrafast radical-mediated pathways and comparatively slow thermodynamic relaxation processes. Consequently, network design should be regarded not only as an implementation detail, but as a fundamental determinant of whether a model can faithfully reproduce combustion phenomena across a broad spectrum of operating conditions.

Among the emerging operator-learning paradigms, DeepONet has attracted substantial attention. By decomposing the mapping into a branch network (encoding input functions) and a trunk network (encoding target coordinates), DeepONet has demonstrated considerable success in learning nonlinear input–output operators, making it attractive for stiff ODEs and parametric PDEs (Lu et al., 2021; Wang et al., 2021; Li et al., 2020). Nevertheless, most existing studies have focused on relatively simplified scenarios—such as narrow ranges of boundary or initial conditions, reduced-dimensional systems, or short integration horizons—which restricts their direct applicability to realistic combustion environments. In many studies with DeepONet, the training datasets and temporal discretization are chosen in an artificial way, not reflecting realistic combustion conditions. For example, in Goswami et al. (2024), the syngas dataset was generated at a fixed chemistry timestep ($\Delta t_{\mathrm{chm}} = 10^{-8}$ s) and the model was trained only to predict four pre-selected future instants ($t_i + 250, 500, 750, 1000$), rather than learning dynamics at arbitrary temporal resolutions. Such limitations arise because the branch–trunk decomposition tends to smooth operator mappings, whereas combustion dynamics often exhibit strongly localized nonlinearities and discontinuities.

This suggests a fundamental open question: can operator-learning architectures such as DeepONet provide superior accuracy and robustness compared to conventional hierarchical models (e.g., U-Net–style residual networks) when applied to reactive flow simulations that more closely approximate real-world scenarios, characterized by extreme transients, multiscale coupling, and broad parameter variations. Addressing this question forms the motivation for the present comparative investigation, in which we systematically evaluate three representative neural architectures—a plain multilayer perceptron (MLP), our custom-designed U-Net–inspired residual network, and a DeepONet-style operator-learning model—on the same high-dimensional dataset.

## 2    PROBLEM STATEMENT

In the present work, the thermal explosion of a hydrogen-oxygen mixture in air is modeled. Having set the initial pressure and temperature, the combustion of the mixture is calculated at regular intervals. The reduced kinetic mechanism proposed in (Tereza et al., 2019) is used to calculate the chemical transformations. During the mechanism 9 hydrogen-oxygen compounds are formed (H2, O2, H2O, OH, H, O, HO2, H2O2, OH*), as well as nitrogen and argon, which affect the dynamics of gases and reaction rates, but do not form compounds, their concentration is a constant value. Chemical kinetics describes the reactions of decay and association of chemical elements at every point in space, described by a system of differential equations:

$$\frac{\partial \mathbf{X}}{\partial t} = f\left(p, T, \mathbf{X}\right) \tag{1}$$

where p is the pressure, T is the temperature, X is the vector of molar densities of elements. Computation of system (1) by numerical methods takes about 90 percent of time resources. The use of neural network models makes it possible to significantly speed up the process of data generation and

calculations, which is important for practical application of combustion and detonation calculations. At the same time, high speed should not reduce the accuracy of predictions, since the physical validity and suitability of the model depend on it. Enhancing predictive accuracy can involve several strategies, including expanding and diversifying the training dataset. Nevertheless, the architecture of the neural network remains the primary determinant of performance, and this study focuses on elucidating its impact.

## 3    DATA FOR TRAINING AND TESTING OF NEURAL NETWORK

The governing equation (1) possesses a unique solution that can be obtained numerically using the stiff ODE solver developed by (Novikov, 2007). This model describes the autoignition process of a preheated hydrogen-air mixture, characterized by a rapid transition to combustion under nearly adiabatic conditions in a confined volume. A key feature of the numerical algorithm is the semi-analytical computation of the Jacobian matrix of the system's right-hand side, which significantly improves computational efficiency and ensures numerical stability when dealing with strongly stiff kinetics.

Direct simulation of the complete gas-dynamic evolution inside the combustion chamber is computationally extremely demanding, even if simplified two-dimensional models are employed. To mitigate this difficulty while preserving accuracy, the computational domain is discretized into uniform volumetric cells. Chemical kinetics are then solved independently within each cell, provided with a set of initial conditions (pressure, temperature, and molar densities of chemical species). The stiff ODE solver computes the temporal evolution of the chemical system efficiently within each cell.

For the present investigation, this solver was used to generate a large database of chemical states at discrete time intervals under a wide variety of randomized thermodynamic conditions. The sampling space was designed to cover practically relevant ranges of parameters:

$$T \in [250, \ 5000] \ \text{K},$$
$$p \in [10^4, \ 2 \times 10^7] \ \text{Pa},$$
$$\Delta t \in [10^{-10}, \ 10^{-5}] \ \text{s}.$$

This strategy ensured that extreme combustion regimes were included, in terms of capturing the full diversity of chemical behavior: from slow reaction zones to sudden autoignition and explosive events.

The resulting dataset provides a classical reference collection of kinetic trajectories. The generated data (see Fig. 1) were represented as 13-dimensional vectors, each comprising the time step $\Delta t$, the system temperature, and the molar concentrations of 11 chemical species.

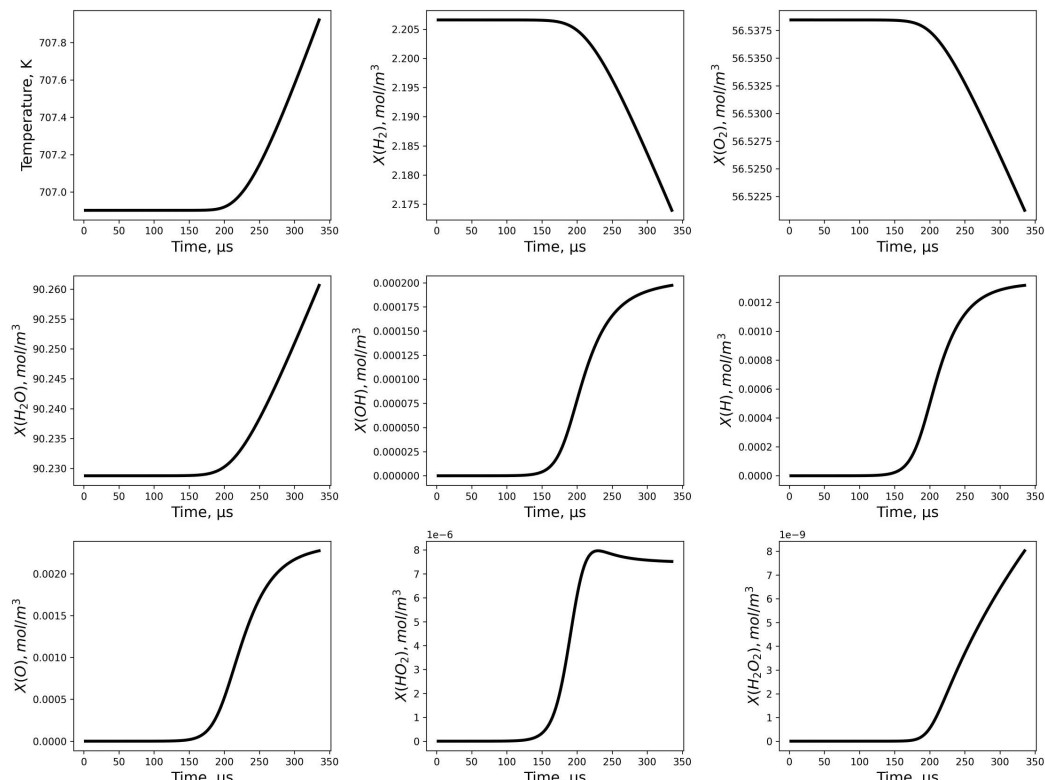

Figure 1: Sample kinetic trajectories from Dataset showing temporal evolution of concentrations and temperature.

Both the input and the output of the neural network therefore had dimension 13. The dataset is split into 50,000 training, 15,000 validation, 5,000 test samples. Dataset represents a broad and relatively unbiased sampling of the chemical system, containing numerous examples of trajectories with long induction times, abrupt ignition events, and smooth transitions to equilibrium. It serves as a general-purpose training dataset for building machine learning models of combustion dynamics.

## 4 NEURAL NETWORK ARCHITECTURES

Figure 2 illustrates the three neural network architectures explored in this work: (A) a plain multi-layer perceptron (MLP), (B) a U-Net–like residual network, and (C) a DeepONet–style model. All three accept the same 13-dimensional input vector

$$X = \big(dt,\ T,\ C_1, \ldots, C_{11}\big),$$

where $dt$ is the time increment, $T$ the temperature, and $C_1 \ldots C_{11}$ the species concentrations (the last two being $N_2$ and Ar). Despite this common input and the shared training procedure, each architecture has distinctive structural features.

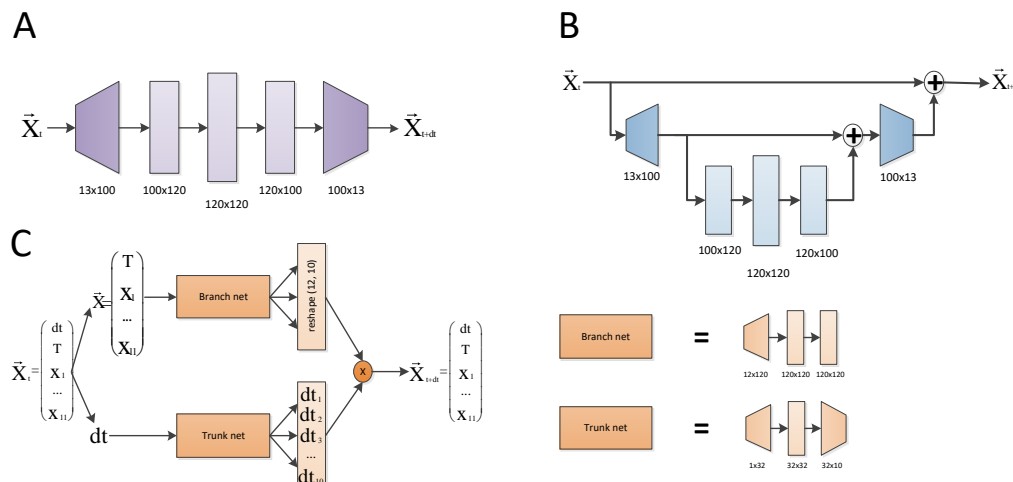

Figure 2: Neural network architectures: (A) plain MLP, (B) U-Net–like residual network, (C) Deep-ONet–style model.

### 4.1 PLAIN MLP

The simplest model is a deep fully connected network with five dense layers: input $13 \times 100 \rightarrow 100 \times 120 \rightarrow 120 \times 120 \rightarrow 120 \times 100 \rightarrow 100 \times 13$ output. A Leaky ReLU activation with negative slope $10^{-2}$ follows every hidden layer. The first and last layers perform dimensional expansion and reduction, while the middle layers provide nonlinear mixing of features. To preserve physically invariant quantities, the output components for $dt$ and the last two concentrations ($N_2$, Ar) are directly copied from the input.

### 4.2 U-NET–LIKE RESIDUAL NETWORK

This architecture adds hierarchical skip connections. The 13-dimensional input is first expanded through a $13 \times 100$ layer, then passed through three dense blocks: $100 \times 120 \rightarrow 120 \times 120 \rightarrow 120 \times 100$ with Leaky ReLU activations. The block output is added to the expansion output (a local skip), followed by a compression layer ($100 \rightarrow 13$) and a global skip that adds the original input vector to the final output. As in the MLP, $dt$ and $N_2$/Ar are enforced to match the input, and the output is clamped to the range $[-10, 10]$.

### 4.3 DEEPONET–STYLE MODEL

This network separates processing of the scalar $dt$ from the remaining 12 variables. One branch maps the 12 state variables through layers $12 \times 120 \rightarrow 120 \times 120 \rightarrow 120 \times 120$, reshaping the result to a $12 \times 10$ matrix. The second branch maps $dt$ through $1 \times 32 \rightarrow 32 \times 32 \rightarrow 32 \times 10$. A matrix product of these branch outputs yields a 12-component fused vector, which is concatenated with $dt$ to form the 13-dimensional output. The $dt$ and $N_2$/Ar components are again fixed to their input values. This design follows the operator-learning principle of DeepONet, where the trunk network (for $dt$) provides coefficients for the basis generated by the branch network.

### 4.4 TRAINING DETAILS

Each layer in all models is a linear transformation

$$h_l = W_l h_{l-1} + b_l, \tag{2}$$

where $W_l$ and $b_l$ are trainable weights and biases. The hidden activations are Leaky ReLU,

$$\text{LeakyReLU}(z) = \max\{0.01z,\ z\}. \tag{3}$$

All models are trained with the Adam optimizer using a learning rate of $0.001$, a batch size of $5,000$, and were trained for 100 epochs. During training, each network minimizes the multi-step prediction error by recursively forecasting the state vector up to thirty steps ahead:

$$\text{Loss} = \sum_{k=1}^{n_{\text{steps}}} \frac{1}{k} \text{MSE}\big(X_{t+k\Delta t},\ \hat{X}_{t+k\Delta t}\big), \tag{4}$$

where $n_{\text{steps}} = 30$. This strategy encourages the models to account for error accumulation while retaining their architectural differences.

## 5 RESULTS

A systematic comparison of three neural network architectures shows how model design directly affects the accuracy of data-driven combustion-dynamics approximation. Performance was quantified using the mean squared error (MSE) on an identical test set to ensure fairness of evaluation. Statistical metrics for all three models are summarized in Table 1. The table includes not only the mean MSE and the standard deviation, but also the 95% confidence intervals (CI) for each model under identical testing conditions.

Table 1: Comparison of MSE statistics for the three architectures on the identical test set

| Model | Mean MSE | Std. Dev. | 95% CI |
|---|---|---|---|
| MLP | $2.029 \times 10^{-2}$ | $6.829 \times 10^{-2}$ | $[1.840 \times 10^{-2}, 2.218 \times 10^{-2}]$ |
| **U-Net** | $\mathbf{1.374 \times 10^{-3}}$ | $\mathbf{2.183 \times 10^{-2}}$ | $\mathbf{[7.692 \times 10^{-4}, 1.980 \times 10^{-3}]}$ |
| DeepONet | $1.808 \times 10^{-2}$ | $5.812 \times 10^{-2}$ | $[1.647 \times 10^{-2}, 1.969 \times 10^{-2}]$ |

The comparatively large spread of errors for all three networks indicates that certain test trajectories remain challenging to approximate. Combustion dynamics often involve abrupt temperature rises, nonlinear chemical–kinetic feedback, and multi-scale temporal behavior. These features can produce regimes that are difficult for purely data-driven models to capture accurately, even when the training dataset is extensive and diverse.

Despite identical training conditions and data preprocessing, the difference in MSE demonstrates that network architecture is an important factor. Direct comparison of the 95% CIs shows that the U-Net's interval $[7.692 \times 10^{-4}, 1.980 \times 10^{-3}]$ does not overlap with the intervals of either MLP or DeepONet, confirming a statistically significant improvement. In contrast, the DeepONet and MLP intervals $[1.647 \times 10^{-2}, 1.969 \times 10^{-2}]$ and $[1.840 \times 10^{-2}, 2.218 \times 10^{-2}]$ overlap substantially, suggesting comparable performance between these two architectures within the test conditions.

The standard deviation of U-Net ($2.183 \times 10^{-2}$) is also much smaller in absolute terms than that of MLP ($6.829 \times 10^{-2}$) and DeepONet ($5.812 \times 10^{-2}$), indicating more stable predictions across diverse trajectories. The U-Net's encoder–decoder design with skip connections appears to capture both global trends and localized transients without increasing computational cost relative to the simpler models. This multi-scale representation likely underlies its lower MSE, narrower CI, and reduced variability, making it the most reliable architecture among those tested.

A detailed analysis of individual test cases further supports this conclusion. All trajectories in the figures (Figure 3 and Figure 4) are plotted in the same normalized space that was used to train the networks. Thus, the vertical axis in each subplot represents the dimensionless normalized value ensuring direct comparability of predicted and reference profiles.

Figure 3 illustrates a representative high-quality prediction: this trajectory belongs to the lowest 10% of test-sample MSE values (i.e., among the best cases for each model). The U-Net captures both transient peaks and long-term plateaus more accurately than the other models, with its output remaining phase-aligned with the true dynamics—peaks, plateaus, and sharp decays occur at the correct times.

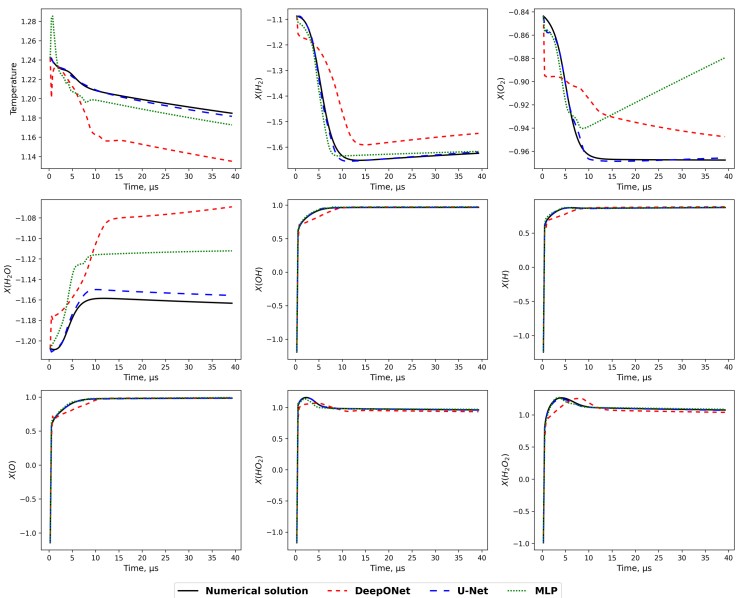

Figure 3: Comparison of predicted time evolution of normalized temperature and normalized concentrations for a representative test trajectory with low MSE. Black solid lines – reference numerical solution; red dashed – DeepONet predictions; blue dashed – U-Net; green dotted – MLP.

By contrast, Figure 4 shows a trajectory from the upper quartile of the MSE distribution, representing a more difficult case. Even in this challenging example, the U-Net output remains aligned with the true temporal trends, preserving ignition peaks and decay phases, whereas the DeepONet and MLP predictions gradually drift away and exhibit phase lag.

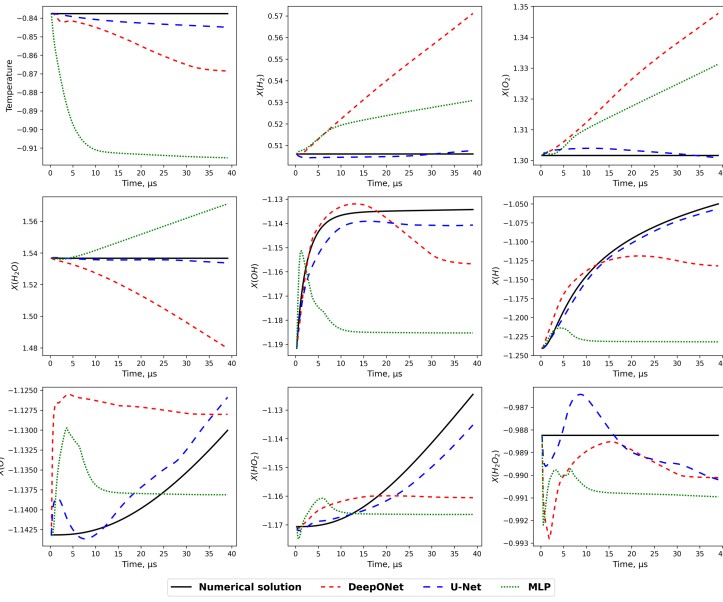

Figure 4: Representative predictions of normalized temperature and key species concentrations for a trajectory with comparatively high MSE. Black solid lines – reference solution; red dashed – DeepONet; blue dashed – U-Net; green dotted – MLP.

These findings emphasize that, for stiff chemical–kinetic systems, incorporating hierarchical feature extraction and residual connections—as in the U-Net—substantially enhances both accuracy and

robustness. Such design choices are as important as dataset size or optimizer tuning and should guide future efforts to create reliable, physics-aware surrogates for combustion simulations.

## 6    CONCLUSIONS

This study demonstrates that neural network architecture plays a key role in the accuracy and robustness of data-driven modeling of reactive kinetics. Among the tested models, the U-Net–style residual network consistently outperformed both the MLP and the DeepONet-inspired architecture, achieving substantially lower mean squared error and reduced variability in predictions. Importantly, this improvement was not limited to absolute accuracy: the U-Net preserved the correct qualitative dynamics of combustion processes, maintaining synchrony with reference trajectories across sharp transients and plateau-like regimes, whereas the other architectures tended to drift away and lose physical consistency.

Our comparative analysis illustrates that the choice of architecture can be as critical as the size or the diversity of the dataset. While MLP and DeepONet-based models captured only partial aspects of the dynamics, the U-Net–style design provided stable and physically meaningful approximations without requiring additional data or computational cost. These findings illustrate the importance of architectural design in constructing reliable neural network surrogates for complex chemical systems.

Overall, the results confirm the promise of U-Net–based architectures and emphasize the potential of combining deep learning with physically motivated design principles to create interpretable, accurate, and robust tools for chemical kinetics.

### ACKNOWLEDGMENTS

Will be added after double-blind review.

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
