# OpenReview forum: "Comparison of Neural Network Architectures in the Thermal Explosion Approximation Problem"
_ICLR.cc/2026/Conference — ICLR 2026 Conference Withdrawn Submission_

### Official Review · Reviewer_Qy6Y · 2025-10-19

**Soundness:** 2
**Presentation:** 2
**Contribution:** 2
**Rating:** 2
**Confidence:** 3

**Summary:**

The paper compares three neural network architectures—MLP, U-Net–like residual, and DeepONet—for modeling the thermochemical evolution of reactive hydrogen–oxygen mixtures. Using a dataset generated from stiff ODE simulations across wide ranges of temperature, pressure, and timestep, each model predicts the temporal evolution of species concentrations and temperature. Despite identical training conditions, results show that the U-Net–like residual network achieves the lowest mean squared error and narrowest confidence interval, indicating superior accuracy and stability compared to MLP and DeepONet. The hierarchical skip connections in the U-Net effectively capture both global and localized transients without increasing computational cost, while DeepONet tends to oversmooth nonlinear dynamics. Overall, the study concludes that for stiff chemical–kinetic systems, architectures incorporating hierarchical feature extraction and residual connections substantially improve predictive accuracy and robustness over conventional feedforward or operator-learning models.

**Strengths:**

The paper presents a systematic comparison of three neural network architectures using a well-defined combustion kinetics dataset, which provides useful benchmarking for engineering and scientific modeling tasks.

The U-Net–like residual network is implemented thoughtfully, with careful consideration of physical invariants and stability, demonstrating solid experimental design and clear reporting.

**Weaknesses:**

The methodological novelty is limited; the architectures (MLP, U-Net, DeepONet) are standard, and the study mainly applies them rather than proposing new theoretical or algorithmic contributions.

The scientific insight gained is incremental, focusing on empirical error comparison without deeper analysis of why the architectures differ in performance (e.g., sensitivity analysis, interpretability, or dynamical reasoning).

Overall, while the work is technically sound, it aligns more with engineering model assessment than with the innovation or conceptual depth expected at a venue like ICLR.

**Questions:**

Have the authors considered extending their comparison to physics-informed neural networks (PINNs) or Fourier-based architectures (e.g., FNOs), which might capture the multiscale coupling and stiffness of chemical kinetics more effectively?

---

> ### Author Response · Authors · 2025-11-27
> **Comment**
>
> Dear Reviewer,
> Thank you for raising important point regarding Physics-Informed Neural Networks (PINNs) and Fourier Neural Operators (FNOs). We agree that these are powerful architectures, and their potential for scientific problems is significant. Our decision not to include them in this specific study was a deliberate:
> 1. On PINNs:
> PINNs are designed to solve Partial Differential Equations (PDEs) by incorporating the governing equations directly into the loss function as soft constraints. However, the problem we address is fundamentally different:
> We are learning a zero-dimensional dynamical system described by a system of Ordinary Differential Equations (ODEs) for chemical kinetics within a single, homogeneous reactor cell. The spatial derivatives that are central to PDEs (and thus to PINNs) are absent in our formulation. The primary difficulty in our problem is not solving the governing equation (which we already do accurately with a stiff ODE solver to generate data) but rather learning a fast and accurate surrogate for the right-hand side (RHS) of these ODEs or their one-step integration map.
> Therefore, while PINNs are excellent for spatial-temporal field reconstruction, they are not a direct competitor for the task of creating a fast, data-driven surrogate for a pre-computed chemical kinetics ODE system, which is the focus of our work.
> 2. On FNOs:
> Our input and output are low-dimensional vectors (13 dimensions), not discretized spatial fields or functions. The powerful spatial convolution performed by FNOs is designed to capture long-range spatial dependencies, which are not present in our zero-dimensional, point-wise chemical kinetics problem.
>
> We sincerely thank you for review. It highlights a direction for future work.

---

### Official Review · Reviewer_sEcu · 2025-10-28

**Soundness:** 2
**Presentation:** 2
**Contribution:** 1
**Rating:** 2
**Confidence:** 5

**Summary:**

This paper studies the effect of neural networks models on modeling of thermal explosions in hydrogen-oxygen-air mixture, simulated under specified initial conditions. Three architectures are compared, an MLP, an adapted DeepONet and a UNet for the ability to capture transient dynamics and reaction regimes. Data is gathered by running an ODE solver for the problem with a range of parameters. The paper concludes that the UNet model consistently outperforms other models with the MLP and DeepONet showing worse performance and greater variability.

**Strengths:**

The paper attempts to provide a comparison of existing neural network architectures for dynamical problems from applied science, which should be of general interest to the community.

**Weaknesses:**

The paper does not propose a new method. There is no novelty either in the experiments, analysis or results.

There is no real analysis of the models employed in terms of identifying limitations of each with regard to performance.

The experimental setup is simplistic and involves training prior models on a single problem and comparing the resulting losses, standard deviations and confidence intervals in Table 1. Figure 3, 4 shows learned trajectories for two cases.

Literature review is sparse and there is no extensive review of the state-of-the-art.

Only three models are compared on a non-standard dataset without making any reference to prior work in comparing neural operator models.

Only a single custom dataset appears to have been used in the experiments. That dataset itself appears to be small with 50k examples of 13d vectors.

**Questions:**

Are there comparisons on more complex problems and would they lead to the same conclusion?

Given that the problem has an ODE formulation have you tried ODE based methods?

---

> ### Author Response · Authors · 2025-11-27
> **Comment**
>
> Dear Reviewer,
> Thank you for your thoughtful critique. Please find below the respond to your major points and outline intended manuscript improvements.
> 1. On novelty and scope.
> We agree the tested architectures are not new. The paper’s contribution is an empirical: a careful, controlled comparison of representative architectures on highly stiff, multi-scale chemical kinetics (the autoignition / thermal explosion problem).
> 2. Dataset size and generalization.
> Although the per-sample dimension is 13, the dataset spans wide ranges of temperature, pressure, and Δt, producing trajectories that include slow kinetics, steep induction phases, and explosive transients. Still, we will add experiments on an extended dataset in our future works.
> 3. On ODE-based methods.
> Neural ODEs belong to a fundamentally different class of continuous-depth models. Their primary goal is often to learn a continuous latent dynamic, and they use an internal ODE solver for inference. In contrast, our work focuses on discrete, single-step surrogate models that are designed to be drop-in replacements for the most expensive part of existing CFD solvers — the ODE integration of chemical kinetics. Our models are meant to be fast, deterministic function approximators, not ODE solvers themselves. Also, neural ODEs introduce significant computational overhead and training instability due to the need for backpropogation through an ODE solver (via the adjoint method). For stiff chemical systems like ours, this often leads to numerical instability and requires very careful tuning of solver tolerances.
>
> Thank you again for your review!

---

### Official Review · Reviewer_d5bQ · 2025-10-28

**Soundness:** 3
**Presentation:** 3
**Contribution:** 1
**Rating:** 2
**Confidence:** 5

**Summary:**

The paper compares three neural network architectures (U-net, DeepOnet and MLP) for learning the solution of an ODE modeling chemical kinetics at play in thermal explosion of a hydrogen-oxygen mixture. The authors find out that a U-net adapted for this low dimensional learning problem yields the best results.

**Strengths:**

- The paper is well written and the scope of the contribution is clearly introduced. As a result, the paper is pleasant to read.
- The methodology is correct

**Weaknesses:**

- The main contribution of the paper is to test several existing architectures on a physical problem. There is no novelty in the types of neural networks introduced, and the learning problem does not seem challenging (13 dimensions, with smooth functions for every dimension), which makes the paper more suited to a journal focused on physical sciences.

- The authors claim that the paper demonstrate that "network architecture has an important impact on predictive performances". This statement is commonly accepted and has been the main motivation for decades of research in deep learning, which makes this claim unsound.

- There are plenty of more modern neural architectures in scientific ML which could be compared to those implemented in the paper.

In the end the paper comes down to a comparison of three architectures on one dataset, which is too light as a contribution for ICLR.

**Questions:**

**Q1** The dimensions of the dataset consists of:

- The temperature

- The time

- The 11 chemical species


But what about the pressure that has been sampled along with the temperature and the time for building the dataset (l.137)

**Q2** How is the time included in the dataset ? For a given sample, what is the value of T for the output given the value of t in the input ? (Is it $t$ and $t + \Delta t$?)

---

> ### Author Response · Authors · 2025-11-27
> **Comment**
>
> Dear Reviewer,
> Thank you for your careful reading and constructive comments. These points are crucial for understanding our methodological choices. Please find our detailed responses below.
> 1. On the role of pressure in the dataset and model.
> You noted that pressure is not part of the 13-dimensional output vector of the network. This is a deliberate design choice grounded in the physics of the problem. Our system is modeled as a zero-dimensional reactor (a single computational cell) with a fixed volume. In this context, the pressure is not an independent state variable, it thermodynamic consequence of the system's composition and temperature. Of course, the initial pressure is a key parameter used to generate the dataset, defining the initial condition for each trajectory. Therefore, the pressure is an implicit function of the state vector.
> By providing the network with the current temperature and concentrations, it has all the necessary information to learn the thermodynamic evolution, including the implicit change in pressure, without requiring it to be an explicit output. This reduces the dimensionality of the learning task.
> 2. On the representation of time.
> We do not use an absolute time coordinate. Instead, we frame the problem as learning a one-step integration operator. Input: (∆t, T(t), C1-11(t)). Output: (∆t, T(t +∆t), C1-11(t+∆t)). The time step ∆t is an explicit input to this operator. This approach has several key advantages:
> Flexibility: The model can be queried with different step sizes without retraining, which is essential for adaptive ODE solvers.
> Focus on Dynamics: It forces the network to learn the local rate of change of the system as a function of the current state and the proposed step size, which is a more fundamental property than memorizing a trajectory in absolute time.
> Stability: It inherently builds a structure that is more amenable to handling the stiffness of the system, as the step size is a direct input.
> Thank you again for your thorough review!

---

### Official Review · Reviewer_nHmd · 2025-10-31

**Soundness:** 2
**Presentation:** 2
**Contribution:** 2
**Rating:** 2
**Confidence:** 3

**Summary:**

This paper presents a comparative study of three different neural network architectures for modeling the thermal explosion process in a hydrogen-oxygen-air mixture. The authors generate a dataset by simulating the chemical kinetics using a reduced 11-reagent mechanism under a wide range of initial conditions. The goal is to predict the temporal evolution of temperature and species concentrations. The architectures compared are a standard Multi-Layer Perceptron (MLP), a DeepONet-inspired model, and a custom U-Net-style residual network. The study finds that the U-Net architecture significantly outperforms the other two, achieving a much lower Mean Squared Error (MSE) and smaller standard deviation in its predictions. The authors conclude that architectural choice is of paramount importance for accurately modeling stiff, multi-scale chemical kinetics, and that hierarchical, residual-based designs like the U-Net are better suited for this task than simpler feed-forward or operator-learning models like DeepONet.

**Strengths:**

1.  **Relevant Problem:** The work tackles the important and challenging task of accelerating stiff chemical kinetics simulations, a key bottleneck in computational combustion.
2.  **Clear Empirical Result:** The paper demonstrates a clear and statistically significant performance advantage of the U-Net architecture over the other two models on the given dataset.
3.  **Well-Generated Dataset:** The authors have created a comprehensive dataset covering a wide range of conditions, including extreme combustion regimes, which is valuable for benchmarking models.

**Weaknesses:**

1.  **Questionable DeepONet Implementation:** The design of the "DeepONet-style model" is non-standard and arguably misrepresents the operator-learning paradigm it is supposed to evaluate. The choice to use the scalar time step `dt` as the input to the trunk network is not well-justified and likely responsible for its poor performance.
2.  **Narrow Architectural Scope:** The comparison is limited to three architectures, omitting many modern and relevant models for sequence modeling, such as Transformers or recurrent networks, which could be strong contenders for this task.
3.  **Lack of Deeper Analysis:** The paper successfully shows *that* U-Net works better, but it does not provide a deep analysis of *why*. There is no investigation into how the multi-scale features learned by the U-Net correspond to the different timescales of the chemical reactions, for instance.
4.  **Conflation of One-Step and Rollout Error:** The analysis does not distinguish between the models' single-step predictive accuracy and their stability in autoregressive rollouts, making it difficult to pinpoint the exact source of the U-Net's superiority.

**Questions:**

1.  Can the authors justify their specific implementation of the DeepONet-style model? Why was the scalar time step `dt` chosen as the input to the trunk network, rather than, for example, a time variable `t` that would be more aligned with learning a time-dependent operator? Have you considered a more standard DeepONet setup where the branch network might encode initial conditions and the trunk network predicts the state at a query time `t`?
2.  Why were other prominent architectures for sequence modeling, such as Transformers or LSTMs, not included in this comparison? Given their success in capturing long-range dependencies in time-series data, they seem like natural candidates for this problem.
3.  Could you provide results for the single-step prediction error (i.e., `n_steps = 1`) for all three models? This would help clarify whether the U-Net's advantage comes from better single-step accuracy, better stability during rollouts, or both.
4.  The paper concludes that the problem of approximating combustion modes "remains unresolved." Could you clarify this statement? Does this refer to the existence of a few high-error trajectories for all models, or a more fundamental limitation? Given the U-Net's strong performance, what specific aspects of the problem do you believe are still unresolved?

---

> ### Author Response · Authors · 2025-11-27
> **Comment**
>
> Dear Reviewer,
> Thank you for your careful reading and constructive comments. Below I respond to your main points.
> 1. On the DeepONet implementation and the choice of dt as the trunk input.
> We agree that our implementation deviates from the canonical DeepONet formulation and appreciate you flagging that. The choice to treat Δt as the trunk input was deliberate: we frame the learning task as the one-step integration operator
> Xt -> Xt+Δt.
> In this formulation the mapping depends explicitly on the integration step size, so we encoded Δt in the trunk to act as the operator’s “query” parameter.
> 2. Why did we not include Transformers / LSTMs / other sequence models?
> We deliberately limited the initial study to architectures with comparable compute/parameter scale to isolate architecture design effects (skip connections, hierarchical features, branch/trunk decomposition).
> 3. Single-step results (n_steps = 1).
> We obtained the single-step errors (as requested):
> 	U-Net: MSE = 0.086, STD = 0.0413
> 	DeepONet: MSE = 0.119, STD = 0.2370
> 	MLP: MSE = 0.4037, STD = 0.3806
> These show that the U-Net advantage is already present at the single-step level; it is not only a rollout stability effect.
> 4. On clarification of  “the problem remains unresolved”.
> We meant that, although the U-Net markedly improves average accuracy and reduces variability, there remains a small but persistent subset of trajectories with high error. These correspond to regimes where stiffness and nonlinearity change fast (very short induction followed by explosive radical growth).
> Thank you again for your review!

---

### Note · Authors · 2025-12-25

I have read and agree with the venue's withdrawal policy on behalf of myself and my co-authors.